# Predicting Parameters in Deep Learning

**Misha Denil**[1]    **Babak Shakibi**[2]    **Laurent Dinh**[3]
**Marc'Aurelio Ranzato**[4]    **Nando de Freitas**[1,2]
[1]University of Oxford, United Kingdom
[2]University of British Columbia, Canada
[3]Université de Montréal, Canada
[4]Facebook Inc., USA
{misha.denil,nando.de.freitas}@cs.ox.ac.uk
laurent.dinh@umontreal.ca
ranzato@fb.com

## Abstract

We demonstrate that there is significant redundancy in the parameterization of several deep learning models. Given only a few weight values for each feature it is possible to accurately predict the remaining values. Moreover, we show that not only can the parameter values be predicted, but many of them need not be learned at all. We train several different architectures by learning only a small number of weights and predicting the rest. In the best case we are able to predict more than 95% of the weights of a network without any drop in accuracy.

## 1   Introduction

Recent work on scaling deep networks has led to the construction of the largest artificial neural networks to date. It is now possible to train networks with tens of millions [13] or even over a billion parameters [7, 16].

The largest networks (i.e. those of Dean *et al.* [7]) are trained using asynchronous SGD. In this framework many copies of the model parameters are distributed over many machines and updated independently. An additional synchronization mechanism coordinates between the machines to ensure that different copies of the same set of parameters do not drift far from each other.

A major drawback of this technique is that training is very inefficient in how it makes use of parallel resources [1]. In the largest networks of Dean *et al.* [7], where the gains from distribution are largest, distributing the model over 81 machines reduces the training time per mini-batch by a factor of 12, and increasing to 128 machines achieves a speedup factor of roughly 14. While these speedups are very significant, there is a clear trend of diminishing returns as the overhead of coordinating between the machines grows. Other approaches to distributed learning of neural networks involve training in batch mode [8], but these methods have not been scaled nearly as far as their online counterparts.

It seems clear that distributed architectures will always be required for extremely large networks; however, as efficiency decreases with greater distribution, it also makes sense to study techniques for learning larger networks on a single machine. If we can reduce the number of parameters which must be learned and communicated over the network of fixed size, then we can reduce the number of machines required to train it, and hence also reduce the overhead of coordination in a distributed framework.

In this work we study techniques for reducing the number of free parameters in neural networks by exploiting the fact that the weights in learned networks tend to be structured. The technique we present is extremely general, and can be applied to a broad range of models. Our technique is also completely orthogonal to the choice of activation function as well as other learning optimizations; it can work alongside other recent advances in neural network training such as dropout [12], rectified units [20] and maxout [9] without modification.



Figure 1: The first column in each block shows four learned features (parameters of a deep model). The second column shows a few parameters chosen at random from the original set in the first column. The third column shows that this random set can be used to predict the remaining parameters. From left to right the blocks are: (1) a convnet trained on STL-10 (2) an MLP trained on MNIST, (3) a convnet trained on CIFAR-10, (4) Reconstruction ICA trained on Hyvärinen's natural image dataset (5) Reconstruction ICA trained on STL-10.

The intuition motivating the techniques in this paper is the well known observation that the first layer features of a neural network trained on natural image patches tend to be globally smooth with local edge features, similar to local Gabor features [6, 13]. Given this structure, representing the value of each pixel in the feature separately is redundant, since it is highly likely that the value of a pixel will be equal to a weighted average of its neighbours. Taking advantage of this type of structure means we do not need to store weights for every input in each feature. This intuition is illustrated in Figures 1 and 2.

The remainder of this paper is dedicated to elaborating on this observation. We describe a general purpose technique for reducing the number of free parameters in neural networks. The core of the technique is based on representing the weight matrix as a low rank product of two smaller matrices. By factoring the weight matrix we are able to directly control the size of the parameterization by controlling the rank of the weight matrix.

Naïve application of this technique is straightforward but tends to reduce performance of the networks. We show that by carefully constructing one of the factors, while learning only the other factor, we can train networks with vastly fewer parameters which achieve the same performance as full networks with the same structure.

The key to constructing a good first factor is exploiting smoothness in the structure of the inputs. When we have prior knowledge of the smoothness structure we expect to see (e.g. in natural images), we can impose this structure directly through the choice of factor. When no such prior knowledge is available we show that it is still possible to make a good data driven choice.

We demonstrate experimentally that our parameter prediction technique is extremely effective. In the best cases we are able to predict more than 95% of the parameters of a network without any drop in predictive accuracy.

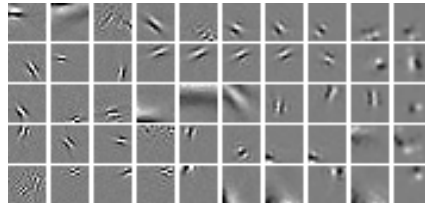

Figure 2: RICA with different amounts of parameter prediction. In the leftmost column 100% of the parameters are learned with L-BFGS. In the rightmost column, only 10% of the parameters learned, while the remaining values are predicted at each iteration. The intermediate columns interpolate between these extremes in increments of 10%.

Throughout this paper we make a distinction between dynamic and static parameters. Dynamic parameters are updated frequently during learning, potentially after each observation or mini-batch. This is in contrast to static parameters, whose values are computed once and not altered. Although the values of these parameters may depend on the data and may be expensive to compute, the computation need only be done once during the entire learning process.

The reason for this distinction is that static parameters are much easier to handle in a distributed system, even if their values must be shared between machines. Since the values of static parameters do not change, access to them does not need to be synchronized. Copies of these parameters can be safely distributed across machines without any of the synchronization overhead incurred by distributing dynamic parameters.

## 2 Low rank weight matrices

Deep networks are composed of several layers of transformations of the form $\mathbf{h} = g(\mathbf{vW})$, where $\mathbf{v}$ is an $n_v$-dimensional input, $\mathbf{h}$ is an $n_h$-dimensional output, and $\mathbf{W}$ is an $n_v \times n_h$ matrix of parameters. A column of $\mathbf{W}$ contains the weights connecting each unit in the visible layer to a single unit in the hidden layer. We can to reduce the number of free parameters by representing $\mathbf{W}$ as the product of two matrices $\mathbf{W} = \mathbf{UV}$, where $\mathbf{U}$ has size $n_v \times n_\alpha$ and $\mathbf{V}$ has size $n_\alpha \times n_h$. By making $n_\alpha$ much smaller than $n_v$ and $n_h$ we achieve a substantial reduction in the number of parameters.

In principle, learning the factored weight matrices is straightforward. We simply replace $\mathbf{W}$ with $\mathbf{UV}$ in the objective function and compute derivatives with respect to $\mathbf{U}$ and $\mathbf{V}$ instead of $\mathbf{W}$. In practice this naïve approach does not preform as well as learning a full rank weight matrix directly.

Moreover, the factored representation has redundancy. If $\mathbf{Q}$ is any invertible matrix of size $n_\alpha \times n_\alpha$ we have $\mathbf{W} = \mathbf{UV} = (\mathbf{UQ})(\mathbf{Q}^{-1}\mathbf{V}) = \tilde{\mathbf{U}}\tilde{\mathbf{V}}$. One way to remove this redundancy is to fix the value of $\mathbf{U}$ and learn only $\mathbf{V}$. The question remains what is a reasonable choice for $\mathbf{U}$? The following section provides an answer to this question.

## 3 Feature prediction

We can exploit the structure in the features of a deep network to represent the features in a much lower dimensional space. To do this we consider the weights connected to a single hidden unit as a function $\mathbf{w} : \mathcal{W} \rightarrow \mathbb{R}$ mapping weight space to real numbers estimate values of this function using regression. In the case of $p \times p$ image patches, $\mathcal{W}$ is the coordinates of each pixel, but other structures for $\mathcal{W}$ are possible.

A simple regression model which is appropriate here is a linear combination of basis functions. In this view the columns of $\mathbf{U}$ form a dictionary of basis functions, and the features of the network are linear combinations of these features parameterized by $\mathbf{V}$. The problem thus becomes one of choosing a good base dictionary for representing network features.

### 3.1 Choice of dictionary

The base dictionary for feature prediction can be constructed in several ways. An obvious choice is to train a single layer unsupervised model and use the features from that model as a dictionary. This approach has the advantage of being extremely flexible—no assumptions about the structure of feature space are required—but has the drawback of requiring an additional training phase.

When we have prior knowledge about the structure of feature space we can exploit it to construct an appropriate dictionary. For example when learning features for images we could choose $\mathbf{U}$ to be a selection of Fourier or wavelet bases to encode our expectation of smoothness.

We can also build $\mathbf{U}$ using kernels that encode prior knowledge. One way to achieve this is via kernel ridge regression [25]. Let $\mathbf{w}_\alpha$ denote the observed values of the weight vector $\mathbf{w}$ on a restricted subset of its domain $\alpha \subset \mathcal{W}$. We introduce a kernel matrix $\mathbf{K}_\alpha$, with entries $(\mathbf{K}_\alpha)_{ij} = k(i,j)$, to model the covariance between locations $i, j \in \alpha$. The parameters at these locations are $(\mathbf{w}_\alpha)_i$ and $(\mathbf{w}_\alpha)_j$. The kernel enables us to make smooth predictions of the parameter vector over the entire domain $\mathcal{W}$ using the standard kernel ridge predictor:

$$\mathbf{w} = \mathbf{k}_\alpha^{\mathrm{T}}(\mathbf{K}_\alpha + \lambda\mathbf{I})^{-1}\mathbf{w}_\alpha \ ,$$

where $\mathbf{k}_\alpha$ is a matrix whose elements are given by $(\mathbf{k}_\alpha)_{ij} = k(i,j)$ for $i \in \alpha$ and $j \in \mathcal{W}$, and $\lambda$ is a ridge regularization coefficient. In this case we have $\mathbf{U} = \mathbf{k}_\alpha^{\mathrm{T}}(\mathbf{K}_\alpha + \lambda\mathbf{I})^{-1}$ and $\mathbf{V} = \mathbf{w}_\alpha$.

### 3.2 A concrete example

In this section we describe the feature prediction process as it applies to features derived from image patches using kernel ridge regression, since the intuition is strongest in this case. We defer a discussion of how to select a kernel for deep layers as well as for non-image data in the visible layer to a later section. In those settings the prediction process is formally identical, but the intuition is less clear.

If $\mathbf{v}$ is a vectorized image patch corresponding to the visible layer of a standard neural network then the hidden activity induced by this patch is given by $\mathbf{h} = g(\mathbf{v}\mathbf{W})$, where $g$ is the network nonlinearity and $\mathbf{W} = [\mathbf{w}_1, \ldots, \mathbf{w}_{n_h}]$ is a weight matrix whose columns each correspond to features which are to be matched to the visible layer.

We consider a single column of the weight matrix, $\mathbf{w}$, whose elements are indexed by $i \in \mathcal{W}$. In the case of an image patch these indices are multidimensional $i = (i_x, i_y, i_c)$, indicating the spatial location and colour channel of the index $i$. We select locations $\alpha \subset \mathcal{W}$ at which to represent the filter explicitly and use $\mathbf{w}_\alpha$ to denote the vector of weights at these locations.

There are a wide variety of options for how $\alpha$ can be selected. We have found that choosing $\alpha$ uniformly at random from $\mathcal{W}$ (but tied across channels) works well; however, it is possible that performance could be improved by carefully designing a process for selecting $\alpha$.

We can use values for $\mathbf{w}_\alpha$ to predict the full feature as $\mathbf{w} = \mathbf{k}_\alpha^{\mathrm{T}}(\mathbf{K}_\alpha + \lambda\mathbf{I})^{-1}\mathbf{w}_\alpha$. Notice that we can predict the entire feature matrix in parallel using $\mathbf{W} = \mathbf{k}_\alpha^{\mathrm{T}}(\mathbf{K}_\alpha + \lambda\mathbf{I})^{-1}\mathbf{W}_\alpha$ where $\mathbf{W}_\alpha = [(\mathbf{w}_1)_\alpha, \ldots, (\mathbf{w}_{n_h})_\alpha]$.

For image patches, where we expect smoothness in pixel space, an appropriate kernel is the squared exponential kernel

$$k(i, j) = \exp\left(-\frac{(i_x - j_x)^2 + (i_y - j_y)^2}{2\sigma^2}\right)$$

where $\sigma$ is a length scale parameter which controls the degree of smoothness.

Here $\alpha$ has a convenient interpretation as the set of pixel locations in the image, each corresponding to a basis function in the dictionary defined by the kernel. More generically we will use $\alpha$ to index a collection of dictionary elements in the remainder of the paper, even when a dictionary element may not correspond directly to a pixel location as in this example.

## 3.3   Interpretation as pooling

So far we have motivated our technique as a method for predicting features in a neural network; however, the same approach can also be interpreted as a linear pooling process.

Recall that the hidden activations in a standard neural network before applying the nonlinearity are given by $g^{-1}(\mathbf{h}) = \mathbf{v}\mathbf{W}$. Our motivation has proceeded along the lines of replacing $\mathbf{W}$ with $\mathbf{U}_\alpha\mathbf{W}_\alpha$ and discussing the relationship between $\mathbf{W}$ and its predicted counterpart.

Alternatively we can write $g^{-1}(\mathbf{h}) = \mathbf{v}_\alpha\mathbf{W}_\alpha$ where $\mathbf{v}_\alpha = \mathbf{v}\mathbf{U}_\alpha$ is a linear transformation of the data. Under this interpretation we can think of a predicted layer as being composed to two layers internally. The first is a linear layer which applies a fixed pooling operator given by $\mathbf{U}_\alpha$, and the second is an ordinary fully connected layer with $|\alpha|$ visible units.

## 3.4   Columnar architecture

The prediction process we have described so far assumes that $\mathbf{U}_\alpha$ is the same for all features; however, this can be too restrictive. Continuing with the intuition that filters should be smooth local edge detectors we might want to choose $\alpha$ to give high resolution in a local area of pixel space while using a sparser representation in the remainder of the space. Naturally, in this case we would want to choose several different $\alpha$'s, each of which concentrates high resolution information in different regions.

It is straightforward to extend feature prediction to this setting. Suppose we have several different index sets $\alpha_1, \ldots, \alpha_J$ corresponding to elements from a dictionary $\mathbf{U}$. For each $\alpha_j$ we can form the sub-dictionary $\mathbf{U}_{\alpha_j}$ and predicted the feature matrix $\mathbf{W}_j = \mathbf{U}_{\alpha_j}\mathbf{W}_{\alpha_j}$. The full predicted feature matrix is formed by concatenating each of these matrices blockwise $\mathbf{W} = [\mathbf{W}_1, \ldots, \mathbf{W}_J]$. Each block of the full predicted feature matrix can be treated completely independently. Blocks $\mathbf{W}_i$ and $\mathbf{W}_j$ share no parameters—even their corresponding dictionaries are different.

Each $\alpha_j$ can be thought of as defining a column of representation inside the layer. The input to each column is shared, but the representations computed in each column are independent. The output of the layer is obtained by concatenating the output of each column. This is represented graphically in Figure 3.

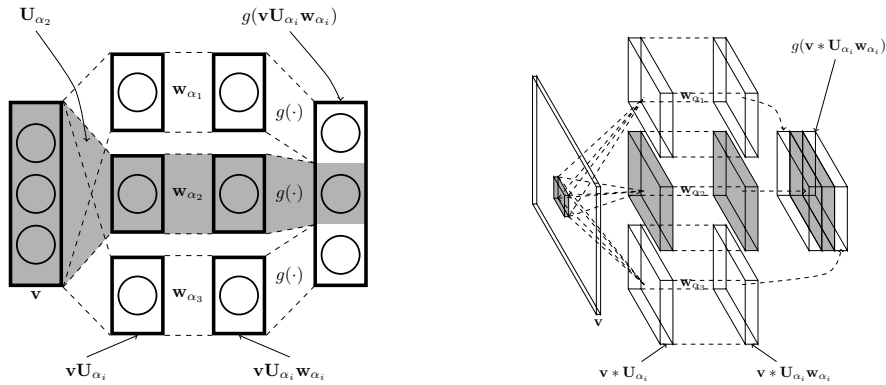

Figure 3: **Left:** Columnar architecture in a fully connected network, with the path through one column highlighted. Each column corresponds to a different $\alpha_j$. **Right:** Columnar architecture in a convolutional network. In this setting the $\mathbf{w}_\alpha$'s take linear combinations of the feature maps obtained by convolving the input with the dictionary. We make the same abuse of notation here as in the main text—the vectorized filter banks must be reshaped before the convolution takes place.

Introducing additional columns into the network increases the number of static parameters but the number of dynamic parameters remains fixed. The increase in static parameters comes from the fact that each column has its own dictionary. The reason that there is not a corresponding increase in the number of dynamic parameters is that for a fixed size hidden layer the hidden units are divided between the columns. The number of dynamic parameters depends only on the number of hidden units and the size of each dictionary.

In a convolutional network the interpretation is similar. In this setting we have $g^{-1}(\mathbf{h}) = \mathbf{v} * \mathbf{W}^*$, where $\mathbf{W}^*$ is an appropriately sized filter bank. Using $\mathbf{W}$ to denote the result of vectorizing the filters of $\mathbf{W}^*$ (as is done in non-convolutional models) we can again write $\mathbf{W} = \mathbf{U}_\alpha \mathbf{w}_\alpha$, and using a slight abuse of notation[1] we can write $g^{-1}(\mathbf{h}) = \mathbf{v} * \mathbf{U}_\alpha \mathbf{w}_\alpha$. As above, we re-order the operations to obtain $g^{-1}(\mathbf{h}) = \mathbf{v}_\alpha \mathbf{w}_\alpha$ resulting in a structure similar to a layer in an ordinary MLP. This structure is illustrated in Figure 3.

Note that $\mathbf{v}$ is first convolved with $\mathbf{U}_\alpha$ to produce $\mathbf{v}_\alpha$. That is, preprocessing in each column comes from a convolution with a *fixed* set of filters, defined by the dictionary. Next, we form linear combinations of these fixed convolutions, with coefficients given by $\mathbf{w}_\alpha$. This particular order of operations may result in computational improvements if the number of hidden channels is larger than $n_\alpha$, or if the elements of $\mathbf{U}_\alpha$ are separable [22].

### 3.5 Constructing dictionaries

We now turn our attention to selecting an appropriate dictionary for different layers of the network. The appropriate choice of dictionary inevitably depends on the structure of the weight space.

When the weight space has a topological structure where we expect smoothness, for example when the weights correspond to pixels in an image patch, we can choose a kernel-based dictionary to enforce the type of smoothness we expect.

When there is no topological structure to exploit, we propose to use data driven dictionaries. An obvious choice here is to use a shallow unsupervised feature learning, such as an autoencoder, to build a dictionary for the layer.

Another option is to construct data-driven kernels for ridge regression. Easy choices here are using the empirical covariance or empirical squared covariance of the hidden units, averaged over the data.

Since the correlations in hidden activities depend on the weights in lower layers we cannot initialize kernels in deep layers in this way without training the previous layers. We handle this by pre-training each layer as an autoencoder. We construct the kernel using the empirical covariance of the hidden units over the data using the pre-trained weights. Once each layer has been pre-trained in this way

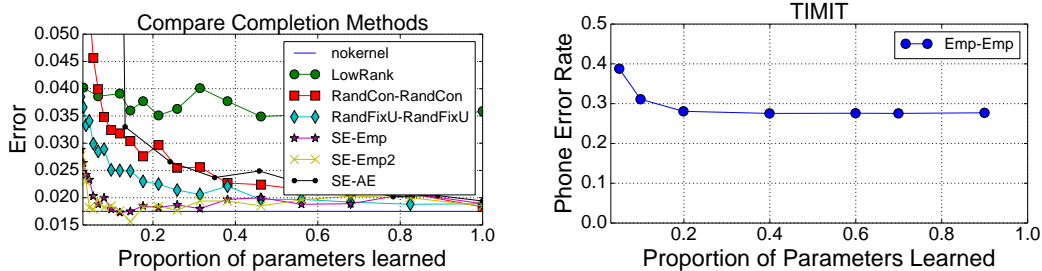

Figure 4: **Left:** Comparing the performance of different dictionaries when predicting the weights in the first two layers of an MLP network on MNIST. The legend shows the dictionary type in layer1–layer2 (see main text for details). **Right:** Performance on the TIMIT core test set using an MLP with two hidden layers.

we fine-tune the entire network with backpropagation, but in this phase the kernel parameters are fixed.

We also experiment with other choices for the dictionary, such as random projections (iid Gaussian dictionary) and random connections (dictionary composed of random columns of the identity).

## 4 Experiments

### 4.1 Multilayer perceptron

We perform some initial experiments using MLPs [24] in order to demonstrate the effectiveness of our technique. We train several MLP models on MNIST using different strategies for constructing the dictionary, different numbers of columns and different degrees of reduction in the number of dynamic parameters used in each feature. We chose to explore these permutations on MNIST since it is small enough to allow us to have broad coverage.

The networks in this experiment all have two hidden layers with a 784–500–500–10 architecture and use a sigmoid activation function. The final layer is a softmax classifier. In all cases we preform parameter prediction in the first and second layers only; the final softmax layer is never predicted. This layer contains approximately 1% of the total network parameters, so a substantial savings is possible even if features in this layer are not predicted.

Figure 4 (left) shows performance using several different strategies for constructing the dictionary, each using 10 columns in the first and second layers. We divide the hidden units in each layer equally between columns (so each column connects to 50 units in the layer above).

The different dictionaries are as follows: *nokernel* is an ordinary model with no feature prediction (shown as a horizontal line). *LowRank* is when both $\mathbf{U}$ and $\mathbf{V}$ are optimized. *RandCon* is random connections (the dictionary is random columns of the identity). *RandFixU* is random projections using a matrix of iid Gaussian entries. *SE* is ridge regression with the squared exponential kernel with length scale 1.0. *Emp* is ridge regression with the covariance kernel. *Emp2* is ridge regression with the squared covariance kernel. *AE* is a dictionary pre-trained as an autoencoder. The *SE–Emp* and *SE-Emp2* architectures preform substantially better than the alternatives, especially with few dynamic parameters.

For consistency we pre-trained all of the models, except for the *LowRank*, as autoencoders. We did not pretrain the *LowRank* model because we found the autoencoder pretraining to be extremely unstable for this model.

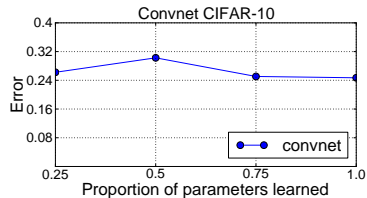

Figure 5: Performance of a convnet on CIFAR-10. Learning only 25% of the parameters has a negligible effect on predictive accuracy.

Figure 4 (right) shows the results of a similar experiment on TIMIT. The raw speech data was analyzed using a 25-ms Hamming window with a 10-ms fixed frame rate. In all the experiments, we represented the speech using 12th-order Mel frequency cepstral coefcients (MFCCs) and energy, along with their first and second temporal derivatives. The networks used in this experiment have two hidden layers with 1024 units. Phone error rate was measured by performing Viterbi decoding

the phones in each utterance using a bigram language model, and confusions between certain sets of phones were ignored as described in [19].

## 4.2 Convolutional network

Figure 5 shows the performance of a convnet [17] on CIFAR-10. The first convolutional layer filters the $32 \times 32 \times 3$ input image using $48$ filters of size $8 \times 8 \times 3$. The second convolutional layer applies $64$ filters of size $8 \times 8 \times 48$ to the output of the first layer. The third convolutional layer further transforms the output of the second layer by applying $64$ filters of size $5 \times 5 \times 64$. The output of the third layer is input to a fully connected layer with $500$ hidden units and finally into a softmax layer with $10$ outputs. Again we do not reduce the parameters in the final softmax layer. The convolutional layers each have one column and the fully connected layer has five columns.

Convolutional layers have a natural topological structure to exploit, so we use an dictionary constructed with the squared exponential kernel in each convolutional layer. The input to the fully connected layer at the top of the network comes from a convolutional layer so we use ridge regression with the squared exponential kernel to predict parameters in this layer as well.

## 4.3 Reconstruction ICA

Reconstruction ICA [15] is a method for learning overcomplete ICA models which is similar to a linear autoencoder network. We demonstrate that we can effectively predict parameters in RICA on both CIFAR-10 and STL-10. In order to use RICA as a classifier we follow the procedure of Coates et al. [6].

Figure 6 (left) shows the results of parameter prediction with RICA on CIFAR-10 and STL-10. RICA is a single layer architecture, and we predict parameters a squared exponential kernel dictionary with a length scale of $1.0$. The *nokernel* line shows the performance of RICA with no feature prediction on the same task. In both cases we are able to predict more than half of the dynamic parameters without a substantial drop in accuracy.

Figure 6 (right) compares the performance of two RICA models with the same number of dynamic parameters. One of the models is ordinary RICA with no parameter prediction and the other has 50% of the parameters in each feature predicted using squared exponential kernel dictionary with a length scale of $1.0$; since 50% of the parameters in each feature are predicted, the second model has twice as many features with the same number of dynamic parameters.

## 5 Related work and future directions

Several other methods for limiting the number of parameters in a neural network have been explored in the literature. An early approach is the technique of "Optimal Brain Damage" [18] which uses approximate second derivative information to remove parameters from an already trained network. This technique does not apply in our setting, since we aim to limit the number of parameters before training, rather than after.

The most common approach to limiting the number of parameters is to use locally connected features [6]. The size of the parameterization of locally connected networks can be further reduced by using tiled convolutional networks [10] in which groups of feature weights which tile the input

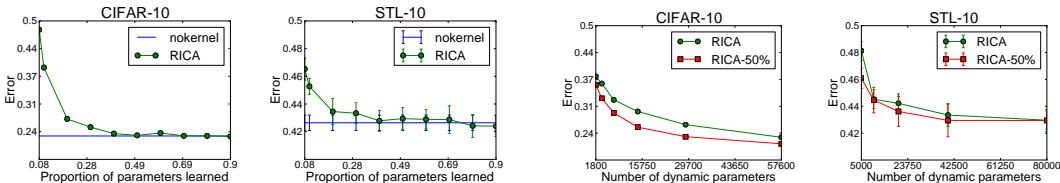

Figure 6: **Left:** Comparison of the performance of RICA with and without parameter prediction on CIFAR-10 and STL-10. **Right:** Comparison of RICA, and RICA with 50% parameter prediction using the same number of dynamic parameters (i.e. RICA-50% has twice as many features). There is a substantial gain in accuracy with the same number of dynamic parameters using our technique. Error bars for STL-10 show 90% confidence intervals from the the recommended testing protocol.

space are tied. Convolutional neural networks [13] are even more restrictive and force a feature to have tied weights for all receptive fields.

Techniques similar to the one in this paper have appeared for shallow models in the computer vision literature. The double sparsity method of Rubinstein et al. [23] involves approximating linear dictionaries with other dictionaries in a similar manner to how we approximate network features. Rigamonti et al. [22] study approximating convolutional filter banks with linear combinations of separable filters. Both of these works focus on shallow single layer models, in contrast to our focus on deep networks.

The techniques described in this paper are orthogonal to the parameter reduction achieved by tying weights in a tiled or convolutional pattern. Tying weights effectively reduces the number of feature maps by constraining features at different locations to share parameters. Our approach reduces the number of parameters required to represent each feature and it is straightforward to incorporate into a tiled or convolutional network.

Cireşan *et al.* [3] control the number of parameters by removing connections between layers in a convolutional network at random. They achieve state-of-the-art results using these randomly connected layers as part of their network. Our technique subsumes the idea of random connections, as described in Section 3.5.

The idea of regularizing networks through prior knowledge of smoothness is not new, but it is a delicate process. Lang and Hinton [14] tried imposing explicit smoothness constraints through regularization but found it to universally reduce performance. Naïvely factoring the weight matrix and learning both factors tends to reduce performance as well. Although the idea is simple conceptually, execution is difficult. Gülçehre et al. [11] have demonstrated that prior knowledge is extremely important during learning, which highlights the importance of introducing it effectively.

Recent work has shown that state of the art results on several benchmark tasks in computer vision can be achieved by training neural networks with several columns of representation [2, 13]. The use of different preprocessing for different columns of representation is of particular relevance [2]. Our approach has an interpretation similar to this as described in Section 3.4. Unlike the work of [2], we do not consider deep columns in this paper; however, collimation is an attractive way for increasing parallelism within a network, as the columns operate completely independently. There is no reason we could not incorporate deeper columns into our networks, and this would make for a potentially interesting avenue of future work.

Our approach is superficially similar to the factored RBM [21, 26], whose parameters form a 3-tensor. Since the total number of parameters in this model is prohibitively large, the tensor is represented as an outer product of three matrices. Major differences between our technique and the factored RBM include the fact that the factored RBM is a specific model, whereas our technique can be applied more broadly—even to factored RBMs. In addition, in a factored RBM all factors are learned, whereas in our approach the dictionary is fixed judiciously.

In this paper we always choose the set $\alpha$ of indices uniformly at random. There are a wide variety of other options which could be considered here. Other works have focused on learning receptive fields directly [5], and would be interesting to incorporate with our technique.

In a similar vein, more careful attention to the selection of kernel functions is appropriate. We have considered some simple examples and shown that they preform well, but our study is hardly exhaustive. Using different types of kernels to encode different types of prior knowledge on the weight space, or even learning the kernel functions directly as part of the optimization procedure as in [27] are possibilities that deserve exploration.

When no natural topology on the weight space is available we infer a topology for the dictionary from empirical statistics; however, it may be possible to instead construct the dictionary to induce a desired topology on the weight space directly. This has parallels to other work on inducing topology in representations [10] as well as work on learning pooling structures in deep networks [4].

## 6 Conclusion

We have shown how to achieve significant reductions in the number of dynamic parameters in deep models. The idea is orthogonal but complementary to recent advances in deep learning, such as dropout, rectified units and maxout. It creates many avenues for future work, such as improving large scale industrial implementations of deep networks, but also brings into question whether we have the right parameterizations in deep learning.

## Footnotes

[1]The vectorized filter bank $\mathbf{W} = \mathbf{U}_\alpha \mathbf{w}_\alpha$ must be reshaped before the convolution takes place.

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
