[Reviews · NeurIPS 2013]

Submitted by Assigned_Reviewer_2

Summary:

"Predicting Parameters in Deep Learning" explores the hypothesis that there is
significant redundancy in the naive (and universally used) representation of
the parameters of neural networks.

Pro:
* Interesting observation and interpretation
* Experiments show promising evidence

Con:
* Empirical results do not support central claims
* Missing comparisons with common weight matrix factoring like PCA preprocessing


Quality:

At a few points in the paper the authors remark on how difficult it is to
train with a parameter matrix W factored as a product UV. Could the authors
offer some intuition for why? One reason might be that if U = V = 0 at the outset,
absolutely naive backprop will not move them. On the other hand if W is
trained normally but simply constrained to be a product of some U and V, some
computational overhead during updates can be traded for lower communication
cost.

A more fundamental concern is that the paper's central observation that
weights are redundant, together with the use of a spatial exponential kernel
suggest that similar gains could be had by either doing a PCA of the input, or
downsampling the images and the kernels for convolution. These are both common
practice. PCA pre-processing also represents a low-rank factorization of the
first layer's weights.

The technique of maintaining a PCA of intermediate layers is not common practice, but
it might be helpful, and it certainly represents a conceptual starting point for your
work on "data driven expander matrices".

The discussion of columnar architectures and convolutional nets seems
reasonable, but largely hypothetical. Pointing out various ways to
refactor matrices in various architectures without more extensive empirical
follow-through seems like a departure from the core thesis of the paper.

The empirical aspect of this paper is important, and I think it needs more
work. The claim that the authors test is that "there exists models whose
parameters can be predicted", but that is a weak claim. The claim the authors are aiming for
from the tone of their introduction and which I agree they *should* be aiming for is that:
"there exist large models that work better than naively down-sized versions
and using our techniques we do less damage to the large models than simply
down-sizing them". This second point is the central claim of the text of the
paper, but it is not supported by empirical results.

The claim that parallel training can actually be accelerated by the use of
these techniques also requires more justification. The proposed method
introduces a few new sorts of computational overhead. Particular
encoder/decoder algorithms for transmitting and rebuilding weight matrices
would need to be implemented and tested to complete this argument.


Clarity:

"collimation" - mis-used word; it is actually a word, but not about making columns

"the the"

"redundancy" is probably the wrong term to describe the symmetry / equivalence
set around line 115.

Section 2 "Low rank weight matrices" is a short bit of background material, it
should probably be merged into the introduction.

Figure 1 is only referenced in the introduction, but it actually illustrates
the method of section 3.2 in action right? It would be clearer if Figure
1 were put into Section 3.2.

Please number your equations, if only for the sake of your reviewers.


Originality:

The idea of looking at weight matrices in neural networks as a continuous
function has precedent, as does the observation that weights have redundancy,
but the idea that training could be accelerated by communicating a small fraction of
randomly chosen matrix values is new.


Significance:

As the authors point out in their introduction, this work could be very
interesting to researchers trying to parallelize neural network training across
very low-bandwidth channels.


Edit: After reading the authors' rebuttal I have raised my quality score. I apparently did not understand the discussion of columns, and how it related to the experimental work. I hope the authors' defense of their experimental work is included into future revisions.
Summary: The paper has some good ideas and gets you thinking, but the empirical results
do not really support the most important and interesting claims. The algorithm
for actually accelerating parallel training is only sketched.

Submitted by Assigned_Reviewer_6

The paper presented a reduction approach to reducing the size of deep models, in order to improve the training efficiency of deep learning. The main contribution is: (1) the exploration of prior knowledge for the redundancy of deep models, such as spatial parameter smoothness for images; (2) the use of kernel ridge regression for parameter interpolation from a subset;

The paper is clearly written. The work seems to be original.

The authors claimed the approach should be very general, i.e., even applicable to non-image tasks. They described some extension of the methods to handle non-image data, such as autoencoder pertaining. But in the experiments there is nothing for non-image datasets. Therefore the point is very weak.

All the experiments on images are on pretty small datasets with simpler patterns. It's hard to believe any methods that are good for data like CIFAR will be supposed good for more realistic datasets such as ImageNet. Therefore the value of this work is not entirely convincing.

I have read the authors' rebuttal. I don't think I would change my recommendation.
Summary: This work is novel, with some quite interesting ideas to reduce the size of deep networks. The value of the work has not been convincingly demonstrated in the experiments.

Submitted by Assigned_Reviewer_7

Motivated by recent attempts to learn very large networks this work proposes an approach for reducing the number of free parameters in neural-network type architectures. The method is based on the intuition that there is typically strong redundancy in the learned parameters (for instance, the first layer filters of of NNs applied to images are smooth): The authors suggest to learn only a subset of the parameter values and to then predicted the remaining ones through some form of interpolation. The proposed approach is evaluated for several architectures (MLP, convolutional NN, reconstruction-ICA) and different vision datasets (MNIST, CIFAR, STL-10). The results suggest that in general it is sufficient to learn fewer than 50% of the parameters without any loss in performance (significantly fewer parameters seem sufficient for MNIST).

The paper is clear and easy to follow. The method is relatively simple: The authors assume a low-rank decomposition of the weight matrix and then further fix one of the two matrices using prior knowledge about the data (e.g., in the vision case, exploiting the fact that nearby pixels - and weights - tend to be correlated). This can be interpreted as predicting the "unobserved" parameters from the subset of learned filter weights via kernel ridge regression, where the kernel captures prior knowledge about the topology / "smoothness" of the weights. For the situation when such prior knowledge is not available the authors describe a way to learn a suitable kernel from data.

The idea of reducing the number of parameters in NN-like architectures through connectivity constraints in itself is of course not novel, and the authors provide a pretty good discussion of related work in section 5. Their method is very closely related to the idea of factorizing weight matrices as is, for instance, commonly done for 3-way RBMs (e.g. ref [22] in the paper), but also occasionally for standard RBMs (e.g. [R1], missing in the paper). The present papers differs from these in that the authors propose to exploit prior knowledge to constrain one of the matrices. As also discussed by the authors, the approach can further be interpreted as a particular type of pooling -- a strategy commonly employed in convolutional neural networks. Another view of the proposed approach is that the filters are represented as a linear combination of basis functions (in the paper, the particular form of the basis functions is determined by the choice of kernel). Such representations have been explored in various forms and to various ends in the computer vision and signal processing literature (see e.g. [R2,R3,R4,R5]). [R4,R5], for instance, represent filters in terms of a linear combination of basis functions that reduce the computational complexity of the filtering process).

In the experimental section the authors make an effort to demonstrate that the proposed method is practically useful and widely applicable, considering multiple datasets and several different architectures. I think, however, that this section could have been stronger in several ways:

Firstly, it would have been nice if the paper had not focused exclusively on vision applications, and had put generally more emphasis on scenarios where there is a less obvious topolgy in the data that can be exploited when predicting weights (which will pose more of a challenge to the method). The data-dependent kernel is not very well evaluated.

Secondly, especially for the vision case, I am wondering why the authors are limiting themselves to the particular set of basis functions derived from the kernel regression view. It would seem natural to explore other linear basis representations of the filters (a simple choice would be a PCA basis, for instance). These might be more efficient in terms of reducing the number of free parameters, and might have other desirable properties (e.g. [R4,R5]).

Finally, I would have hoped for a truly compelling experimental use case demonstrating the impact of the approach in practice. Since the work is motivated as a way of reducing computational and memory complexity, I think it would have been useful to include a more detailed discussion and empirical demonstration how the reduction in number of learned parameters translates into such savings and consequently allows the training of larger networks that achieve better performance than otherwise possible. At the moment, the evaluation appears to be focused on moderately large networks, and the authors show that a moderate reduction in parameters can be achieved that way, without loosing performance (for MNIST the potential reduction in parameters seems to be very large, but for the more interesting CIFAR / STL-10 datasets it seems that at least 40% of the parameters are required to avoid a loss in accuracy). Why is computational complexity and speed of convergence not considered at all in the evaluation? I think that Fig. 6-right (which plots performance against # of free parameters for a standard network and a reduced parameterization) goes in the right direction, but it seems that there is really only a clear advantage of the reduced parameterization for CIFAR (much less so for STL), and I'm wondering whether the performance difference on CIFAR would disappear even for currently realistic network sizes.

All in all, I think the paper takes an interesting perspective although related approaches have appeared in the literature in various guises (see discussion above). I think that the ideas can have practical impact, and to my knowledge, they are currently not widely used in the NN/deep learning community. A stronger case could probably be made, however, by further exploring alternative implementations of the idea and by having a more compelling demonstration of the effectiveness and versatility of the approach.

Further remarks:

** It would have been really helpful to include state of the art results from the literature for the datasets / and specific evaluation regimes considered. This would put the reported performance into perspective and make it easier to judge whether savings in learned parameters can be achieved for competitive architectures.

** I find the evaluation of the data-dependent kernel somewhat limited: it is only explored for a single dataset / architecture combination (the MNIST, MLP experiments in section 4.1). As mentioned above, it would have been nice to inlcude some other type of data that requires the use of the empirical kernel.

For the experiments in section 4.1 it would have further been useful to also consider the case SE-rand. At the moment it's not clear that the learned kernel really outperforms the random approaches. The difference might simply be due to the fact that for the empirical kernel you're using the SE kernel in the first layer. Another interesting control for the effectiveness of the empirical kernel would be to have a setting where you're using the empirical kernel in both layers.

** Do you have any insights as to why the naive low-rank scheme, where both matrices are learned, is working so poorly? Have you made any attempt to remove the redundancy inherent in the parameterization? Would it be possible (and potentially advantageous) to start of with a fixed U (e.g. the empirical kernel), but allow U to change later at some point, possibly in an optimization scheme where U and V are updated in an alternating manner?

** In your conv-net experiments I suppose you are not using any pooling layers as described in [18]? This would also reduce the dimensionality of the representation in the higher layers. Furthermore, based on Fig. 5 I am wondering whether you can really argue that predicting 75% of the parameters has negligible impact on performance: when you're predicting 50% of the parameters the loss seems to be about 5-6 percentage points which does not seem so negligible to me (I guess it would help to have some errorbars here).

** How did you optimize the hyperparameters (learning rate etc.) for your comparisons? Shouldn't the kernel width vary with the size of the subset of learned parameters alpha?

** In section 4.1, for the MLP experiments, how do the columns differ? Do they use the same empirical kernel and differ only in the set of sampled pixels? Is the great advantage of using multiple columns here due to the fact that otherwise you don't sample the space well?

** Fig. 6 right: I find it curious that the reduced parameterization is advantageous for for CIFAR but there seems to be very little difference for STL - given the higher resolution of STL wouldn't one expect the opposite?

** For conv-nets where, in the higher-layers, there are many input channels relative to the spatial extent of each filter, would it be possible to use a data-dependent kernel to subsample across channels (if I understand correctly you are currently performing the subsampling only spatially, i.e. the elements of alphas are tied across channels)?

** Is there a way to extend your scheme such as to start off with a small fraction of learned parameters, but to then increase the fraction of learned parameters slowly until a desired performance is reached / no further improvement is observed?



References:


[R1] Dahl et al., 2012: Training Restricted Boltzmann Machines on Word Observations

[R2] Roth & Black, IJCV, 2009: Fields of Experts (Section 5.3)

[R3] Freeman & Adelson, 1991: The design and use of steerable filters

[R4] Rubinstein et al., 2010: Double Sparsity: Learning Sparse Dictionaries for
Sparse Signal Approximation

[R5] Rigamonti et al., 2013: Learning Separable Filters
Summary: The paper proposes a way of reducing free parameters in NN and I think this can be a useful practical contribution. There is, however, some similarity to existing work, and a stronger case for the approach could probably be made by considering alternative implementations and providing a more compelling evaluation.

Author Feedback

Author rebuttal: This paper proposed a method to reduce the number of parameters of existing DEEP learning approaches. The proposed idea is independent of the specific choice of model and optimization algorithm. Our experiments are aimed at supporting this claim by showing that the idea works for many architectures (RICA, convnets, MLPs) and for the various optimization algorithms associated with each. The strengths of this idea are two-fold: generality and simplicity.

Regarding generality, previous approaches (those we cite and R1-R5 from reviewer 7) have explored factorizations of the weight matrix. Crucially, the factorization techniques are very much tied to the specific optimization and model being used. R5 uses two stages of optimization where ALL weights are first computed and then approximated in a second step. R2 uses PCA bases and instead of learning the second factor, it uses a random matrix. R4 requires alternating sparse coding. It is not obvious how to scale these approaches to arbitrary architectures. More importantly, these ideas were proposed for shallow models, whereas our idea is directly applicable to any deep model and corresponding optimizer. We have focused on images, but it is clear that for speech the same structure in the filters can be exploited (in fact, in speech the signal is typically transformed to an image as a pre-processing step). It is also clear that our idea will work for other datasets such as ImageNet because the filters reported in those works have similar structure. It is not clear how well our idea will work for text, but we think success with images is important enough to warrant publication. R1 proposes a factorization specifically tailored to RBMs and to address repeating tokens within a window of text. This idea could prove useful. In conclusion, the provided references are for shallow models and don't exhibit the generality of our approach.

A strength of our paper is that the idea is simple, but it is only obvious in retrospect. If it were obvious it would be widely used, as having fewer parameters reduces storage and communication costs as also pointed out in the excellent paper [R4]. We have researched this idea extensively on the web, presented it at an ICML workshop and have run it by the key leaders in the field of deep learning. Beyond the references they provided us with, which we cite, everyone thus far has agreed that this is a new idea.

Regarding the amount of experimentation, we tested with a broad set of models, optimizers and datasets. The papers we cite, e.g. for RICA, only used a subset of our datasets when published at NIPS. Our results were the product of several months of work by 4 experts dedicated to this problem. We believe the amount of experimentation is very reasonable for a conference publication.

We arrived at this idea using nonparametrics (the kernel ridge regressor is the mean of a Gaussian process over filters). However, we agree that the perspective as a linear, possibly sparse, combination of bases (PCA, Fourier, wavelets, etc) as in eg R2-R5 is also appealing. We thank the reviewers for suggesting this modification to our work, and we will provide comparisons between this approach and the kernel-based approach in the final version of the paper. Another important insight we gained from the reviews, is that we can make our technique even more efficient by exploiting separability as in some of the references provided by reviewer 7.

The central idea, which we hope the reviewers focus on, is one of reducing the number of parameters of most existing deep learning models and associated algorithms. This idea is new and useful.

Specific feedback

Reviewer 2

Factoring W and optimizing both factors is difficult because of the ambiguity we mention at the end of section 2. It may be possible to make this work well, for instance using alternating optimization as suggested by Reviewer 7. This would reduce the simplicity of our technique, since this involves a modification of the training algorithm, rather than merely the parameterization of the model.

The discussion of columns is not hypothetical; in fact it is used throughout our experiments. In Figure 4 we show that 5 columns outperforms 1 column in MLPs, and although we did not include it, the pattern is similar for other models. The improvement in performance from multiple columns is because each column has dynamic parameters in different locations, giving better coverage of the space.

It has been observed in the past that there is redundancy in the weights, but this redundancy has not been previously exploited in deep learning.

Reviewer 7

Including state of the art results from the literature would detract from the main message of our paper. One of the key strengths of our idea is its generality, which we demonstrate through the use of several different models. It seems clear that models tailored to specific benchmarks will outperform more general approaches on those benchmarks. That said, the performances we report are generally competitive with other modern techniques. For example our experiments with RICA achieve better results on STL10 that the original RICA paper [16].

We did experiment with SE-rand in Section 4.1, but found that SE-emp and SE-emp2 perform better. We did not include this permutation due to space constraints.

Network architectures were chosen in various ways: For MLPs we used one of the architectures from [13]. For convnets we tried architectures by hand until we found a good one. For other hyperparameters we used the defaults in pylearn2 for both models. For RICA we used the same architecture as [16] and used cross validation to optimize one parameter at a time.

There are many possibilities for choosing different sampling patterns for the locations of the alphas; we mention some in section 5. Exploring all the permutations here is beyond the scope of this paper.

See also our response to reviewer 2.